# Heat Shock Protein 70 in Cold-Stressed Farm Animals: Implications for Viral Disease Seasonality

**DOI:** 10.3390/microorganisms13081755

**Published:** 2025-07-27

**Authors:** Fanzhi Kong, Xinyue Zhang, Qi Xiao, Huilin Jia, Tengfei Jiang

**Affiliations:** College of Animal Science and Veterinary Medicine, Heilongjiang Bayi Agricultural University, No. 5 Xinfeng Road, Sartu District, Daqing 163319, China; zhangxinyue1@byau.edu.cn (X.Z.); xiaoqi0401@byau.edu.cn (Q.X.); huilinjia@byau.edu.cn (H.J.); tengfeijiang207@byau.edu.cn (T.J.)

**Keywords:** animal cold stress, viral diseases seasonality, heat shock protein 70, PEDV, PRRSV

## Abstract

The seasonal patterns of viral diseases in farm animals present significant challenges to global livestock productivity, with cold stress emerging as a potential modulator of host–pathogen interactions. This review synthesizes current knowledge on the expression dynamics of heat shock protein 70 (HSP70) in farm animals under cold-stress conditions and its potential roles as (1) a viral replication facilitator and (2) an immune response regulator. This review highlights cold-induced HSP70 overexpression in essential organs, as well as its effects on significant virus life cycles, such as porcine epidemic diarrhea virus (PEDV), porcine reproductive and respiratory syndrome virus (PRRSV), and bovine viral diarrhea virus (BVDV), through processes like viral protein chaperoning, replication complex stabilization, and host defense modulation. By integrating insights from thermophysiology, virology, and immunology, we suggest that HSP70 serves as a crucial link between environmental stress and viral disease seasonality. We also discuss translational opportunities targeting HSP70 pathways to break the cycle of seasonal outbreaks, while addressing key knowledge gaps requiring further investigation. This article provides a framework for understanding climate-driven disease patterns and developing seasonally adjusted intervention strategies.

## 1. Introduction

Environmental exposure to deep freezes, heat waves, and other discrete factors can adversely affect animal health. Cold stress is a crucial factor leading to poor growth performance, possible diseases, and increased mortality rates in animals during the wintertime in temperate regions, consequently reducing economic gains [1,2,3]. Additionally, hypothermia can reduce signal transduction efficiency, slow enzymatic reactions, and diminish membrane transportation, resulting in overall inhibition of protein synthesis [4,5]. In contrast, to maintain homeostasis at the cellular level, some proteins are rapidly upregulated to guard against the drastic effects of cold stress in animals thriving in cold regions [6]. Heat Shock Protein 70 kDa (HSP70) is one of the most ubiquitous classes of chaperones, participating in almost every biological process [7,8]. Under physiological conditions, the expression levels of HSP70 are lower, but they rapidly increase in response to cold stress, improving cellular stress tolerance, maintaining normal metabolism, and enhancing viability [9,10]. HSP70 functions include the folding and assembly of newly synthesized polypeptides, encompassing viral polypeptides, refolding of stress-denatured proteins, translocating organellar and secretory proteins across membranes, preventing aggregation and refolding of misfolded proteins, controlling the activity of regulatory proteins, and assisting in the degradation of terminally misfolded proteins [11,12]. Cold stress not only challenges animal physiology and immunological consequences but also influences the epidemiological patterns of viral diseases, potentially shaping their seasonality. This mini-review aims to synthesize current knowledge on the expression dynamics of HSP70 in cold-stressed farm animals and its potential roles in shaping viral disease seasonality. While a comprehensive systematic review encompasses a broad search strategy and exhaustive analysis of the entire available literature, a mini-review, by its nature, offers a focused and concise overview of a specific topic. Therefore, this work is not intended to be an exhaustive compilation of all research in the field. Instead, we selectively highlight key studies and concepts that, in our judgment, are most pertinent to understanding the multifaceted roles of HSP70 in aiding animals to cope with cold stress and how these functions might intersect with viral disease dynamics. Key studies were identified through database searches (PubMed and Google Scholar) using targeted keywords, with preference given to peer-reviewed articles addressing molecular mechanisms, seasonal disease patterns, and translational implications. We acknowledge that this focused approach may not capture the entire breadth of the existing literature, and we identify key knowledge gaps requiring further investigation to provide directions for future research efforts in this area.

## 2. Expression Dynamics of HSP70 in Cold-Stressed Poultry and Livestock

When living organisms are exposed to various stress conditions, the synthesis of most proteins is slowed, but a group of highly conserved proteins known as heat shock proteins (HSPs) are rapidly synthesized [13]. HSPs serve as major molecular chaperones, performing crucial roles in the folding/unfolding and translocation of proteins, as well as in the assembly/disassembly of protein complexes [14]. Among them, HSP70 is prevalent in virtually all compartments of eukaryotic cells and is involved in a broad spectrum of housekeeping and stress-related protein folding activities [15]. Extensive studies have demonstrated that cold stress elevates HSP70 mRNA and protein levels in the heart, spleen, thymus, bursa of Fabricius, midbrain, forebrain, liver, breast muscle, and small intestine of chickens (Table 1). Similarly, cold stress induces increased expression of HSP70 in the myocardial tissue of Japanese quail, with a strong correlation observed between the magnitude of core temperature decline and the increased level of HSP70 in these birds [16]. Furthermore, increased expression of HSP70 has been found to protect the spleen and cecum against cold stress and inflammatory damage in quails [17,18]. Nevertheless, intermittent mild cold acclimation and prolonged cold exposure can lead to a decrease or increase in HSP70 levels in the liver, jejunum, ileum, and spleen of broilers, resulting in improved immune function and anti-stress abilities [19,20,21]. These studies provided intriguing insights into the potential dual role of HSP70 in cold-stress response. While HSP70 clearly has a protective function in acute cold stress, its continued upregulation during acclimation seems to be modulated, likely reflecting a more complex interplay between cellular protection, immune response, and energy balance. Further research is needed to fully understand the complex interplay of HSP70, cold stress, and acclimation in broilers.

Livestock is a crucial food resource for residents of cold regions, such as northern Asia and alpine areas, where agriculture is limited. In these regions, cold stress negatively affects animal welfare, leading to direct economic losses through mortality and morbidity, as well as indirect costs due to reduced weight gain and performance [22]. Recent efforts have been made to uncover the relationship between HSP70 expression levels and cold tolerance in cattle. An increased expression of HSP70 family genes was observed in buffaloes during the winter season [23]. As Kumar et al. (2015) noted, variations in the expression patterns of HSP70 family genes during different seasons might be linked to the better adaptability of Indian zebu cattle to various climatic conditions [24]. Mayengbam et al. (2016) reported that HSP72 mRNA expression increased during cold stress in the peripheral blood mononuclear cells (PBMCs) of Sahiwal cattle [25]. Bhanuprakash et al. (2016) also found significantly higher levels of HSP70 in PBMCs during cold stress in both Sahiwal and Frieswal cattle, with levels being higher in Sahiwal than in Frieswal cattle [26]. Furthermore, Xu et al. (2017) showed that HSP70 was upregulated in the peripheral blood of Chinese Sanhe cattle exposed to severe cold stress, suggesting it as a candidate gene related to severe cold-stress response in cattle [27]. We found that HSP70 nucleotide and protein levels were increased in the duodenum, jejunum, and ileum of piglets after 12 h of cold exposure at 4 °C [28]. The molecular basis for improved resistance to declining ambient temperatures in relatively thermotolerant cattle is not yet fully understood and requires further investigation. Expression dynamics of HSP70 in cold-stressed poultry and livestock are summarized in Table 1.

**Table 1 microorganisms-13-01755-t001:** Upregulated HSP70 levels in cold-stressed poultry and livestock.

Species	Organs	Increased Levels	Potential Functions of HSP70	Reference
Chicken	Heart	mRNA and proteins	Protects heart against cold stress	[29]
	Spleen, thymus, and bursa of Fabricius	mRNA and proteins	Protects immune organs against cold stress	[30]
	Spleen and bursa of Fabricius	mRNA	-	[31]
	Midbrain and forebrain	mRNA	Delays the pathological process of cold stress	[32]
	Liver	mRNA	-	[33]
	Liver,			
	heart, and breast muscle	mRNA	-	[34]
	Jejunum and ileum	mRNA and proteins	Protects cells against damage caused by cold stress	[20]
Quail	Heart	Protein	Biomarker of ambient hazards	[16]
	Cecum	mRNA	Protects cecum against cold stress	[17]
	Spleen	mRNA	Protects spleen against cold stress	[18]
Bovine	Blood	mRNA	-	[23,24]
	Blood	mRNA	-	[24,27]
	Sertoli cells	mRNA and proteins	Protects sertoli cells against cold stress	[35]
	PBMCs	mRNA and proteins	-	[25,26]
Porcine	Duodenum,	mRNA and proteins	Promotes PEDV replication in vivo	[28]
	jejunum, and			
	ileum			

Note: “-” indicates not reported.

## 3. Seasonal Trends of Farm Animal Viral Diseases

Animal viral diseases often exhibit pronounced seasonality, with many outbreaks occurring during specific times of the year. Seasonal trends in animal viral diseases, particularly during cold seasons, can have significant negative impacts on animal husbandry. For instance, respiratory viral infections, including coronaviruses, bovine respiratory syncytial virus (RSV), porcine reproductive and respiratory syndrome virus (PRRSV), and avian influenza viruses (AIVs), tend to peak in the winter months in temperate regions due to cold and dry conditions that enhance virus stability and transmission while weakening host immune responses [36,37,38,39,40,41,42,43,44]. Similarly, African Swine Fever (ASF) shows distinct seasonal patterns, with outbreaks in wild boar peaking in winter and summer, influenced by factors such as human movement and wild boar behavior [45]. The prevalence of beak and feather disease virus (BFDV) in captive psittacine birds is influenced by species and seasonal factors, with higher prevalence observed during the wet season, possibly due to increased humidity and rainfall [46]. In addition, canine distemper virus (CDV) infection in minks, foxes, and raccoon dogs is more common in winter [47]. Porcine enteric coronaviruses, such as transmissible gastroenteritis virus (TGEV), porcine epidemic diarrhea virus (PEDV), porcine deltacoronavirus (PDCoV), and swine acute diarrhea syndrome coronavirus (SADS-CoV), are significant pathogens causing diarrhea outbreaks in pigs and frequently occur during early spring and winter months in temperate regions, peaking between November and March [48,49,50]. We found that cold exposure-induced HSP70 overexpression positively regulates PEDV replication in IPEC-J2 cells [28], which would be helpful for understanding the seasonality of PED epidemics and the development of novel antiviral therapies targeting HSP70 in the future. Overall, the seasonality of animal viral diseases in husbandry is shaped by a complex interplay of environmental conditions, animal behavior, host–pathogen interactions, vector dynamics, and husbandry practices (Figure 1). Understanding these factors is essential for predicting outbreaks and implementing effective control measures, especially during the cold season when certain diseases may persist or peak. By considering the specific factors that drive disease dynamics in different regions and production systems, stakeholders can develop more targeted and effective strategies to mitigate the impact of these diseases on animal health and productivity.

## 4. Potential Roles of HSP70 in Shaping Farm Animal Viral Disease Seasonality

As a matter of fact, epidemics of some viral diseases, such as respiratory or gastrointestinal tract virus infections, recur with marked seasonality, spanning winter to early spring [40,41,51,52,53,54,55]. Despite extensive documentation of the seasonality of viral diseases and curiosity as to their causes, little concrete data is available to indicate why infections peak in the wintertime. HSP70, a crucial stress-induced protein, has been a focus topic in studying the seasonality of viral infections in farm animals. Emerging research suggests that HSP70 can directly or indirectly impact viral infections by boosting pathogen replication in vivo or dampening the host immune response to viral pathogens.

### 4.1. Roles of Hsp70 in Boosting Viral Replication and Infections

HSP70 proteins are molecular chaperones that assist in protein folding, protect cells from stress, and regulate immune responses. Viruses exploit HSP70 to support their own life cycles, using these chaperones for folding viral proteins; stabilizing replication complexes; and enhancing viral entry, replication, assembly, and release [56]. HSP70 can directly interact with viral polymerases and other viral proteins, facilitating efficient viral replication and survival in host cells under stress conditions. Studies demonstrate its proviral roles in post-entry stages of the Tembusu virus (TMUV) life cycle (replication, assembly, and release) [57] and that of rabies virus (transcription, translation, and virion production) [58]. HSP70 family members (Hsc70/Hsp70) interact with Minute virus of canine (MVC) proteins (NS1 and VP2), directly promoting MVC replication [59], while HSP70-NS5A interactions are critical for classical swine fever virus (CSFV) RNA replication [60]. Conversely, pharmacological inhibition of Hsp70 (e.g., by quercetin) suppresses BVDV infection via blocking oxidative stress and ERK signaling [61], underscoring its targetability. Mechanistically, HSP70 facilitates viral replication complexes by binding dsRNA and structural proteins (e.g., IBDV VP2/VP3 [62], PCV2 Cap [63], and PRRSV dsRNA [64]) and chaperoning polymerase subunits (influenza PB1/PB2 [65]). It also mediates cell entry as part of receptor complexes (IBV S-protein [66]) and endocytic trafficking (PEDV S-protein via HSPA5 [67]). Notably, we found that cold-exposure-induced HSP70 could positively regulate PEDV mRNA synthesis and protein expression in vitro, and this has been further validated by two studies [67,68], suggesting environmental modulation of viral susceptibility (Figure 2). We note that most evidence for HSP70’s proviral roles comes from cell culture models. The extrapolation of these findings to whole-animal systems requires caution, as winter disease patterns reflect complex interactions between environmental persistence, host population immunity, and management factors that may outweigh molecular mechanisms.

While HSP70 predominantly exhibits proviral functions across diverse pathogens, emerging evidence reveals context-dependent antiviral roles. For instance, HSP70 negatively regulates rotavirus RF strain replication in Caco-2 cells by limiting structural protein bioavailability [69], and the HSP70-DnaJC7 complex suppresses Fowl Adenovirus (FAdV-4) via autophagy-mediated Hexon degradation [70]. These dichotomous effects underscore HSP70’s pleiotropy in viral infections. Although HSP70 is among the few host factors experimentally linked to viral disease seasonality in farm animals, its precise contribution relative to other determinants (e.g., environmental stressors and host immunity) remains unresolved. Further research should delineate the molecular switches governing HSP70’s dual roles—whether as a viral facilitator or restriction factor—and explore its therapeutic potential in mitigating seasonal outbreaks to enhance livestock health and productivity. In this study, we are focusing on HSP70’s molecular roles, and we acknowledge the fact that winter disease patterns emerge from complex interactions of environmental persistence (temperature and humidity), host factors (immunity and age), and viral evolution, with HSP70 representing one contributing mechanism. The relative contribution of HSP70 to field epidemiology requires further study alongside environmental and population factors. While in vitro data suggest proviral roles, its actual impact likely varies by production system, virus type, and management practices.

### 4.2. Roles of Hsp70 in Regulating Farm Animal Immune Responses

As a quintessential stress-responsive chaperone, HSP70 exhibits remarkable duality in immune regulation of farm animals, functioning as both an immunosuppressor and immunostimulant, particularly under stress conditions, such as heat, cold, or infection. Its immunosuppressive actions involve enhancing the suppressive capacity of regulatory T and B cells, leading to increased production of anti-inflammatory cytokines, such as IL-10 and TGF-β, thereby dampening effector T cell activity and maintaining immune homeostasis [71]. Extracellular HSP70 promotes antiviral defenses by enhancing antigen presentation and type I interferon responses, yet intracellularly, it fine-tunes these responses by competitively inhibiting IRF7 phosphorylation and IFN-α production [72]. Additionally, HSP70 interacts with toll-like receptors (TLR2 and TLR4), activating signaling pathways that balance immune activation and suppression, which helps prevent excessive inflammation and tissue damage, particularly under stress conditions, like heat and oxidative stress, thus supporting overall resilience [73]. Conversely, HSP70 also functions as an endogenous adjuvant that boosts immune responses; it modulates both innate and adaptive immunity by influencing key signaling pathways, such as the NF-κB pathway, which is essential for the transcription of immune response genes [74]. The protein’s context-dependent immunomodulation is exemplified in autoimmune paradigms, where it can paradoxically promote neuroinflammation in experimental autoimmune encephalomyelitis while suppressing excessive cytokine storms in other settings [75]. Such pleiotropy positions HSP70 as a unique therapeutic target, with current exploration focusing on its potential as a physiological immunomodulator capable of attenuating inflammatory diseases without antigen-specific interventions. Future research should elucidate the molecular switches governing HSP70’s immunological duality, particularly in pathogen-challenged environments where its functions may shift between protective and detrimental roles.

## 5. Other Potential Factors in Shaping Farm Animal Viral Disease Seasonality

### 5.1. Environmental Factors in Viral Persistence

The winter season creates a uniquely favorable environment for viral persistence and transmission in livestock production systems through multiple interconnected mechanisms. Temperature plays a fundamental role, with most enveloped viruses (including PRRSV, influenza viruses, and coronaviruses) demonstrating significantly greater stability in cold conditions [40,76]. Laboratory studies show that at temperatures below 10 °C, viral viability in fomites and aerosols can persist for days to weeks compared to just hours under warmer conditions [40,77]. This temperature effect is particularly pronounced for lipid-enveloped viruses, whose membranes maintain greater structural integrity in cold environments. Humidity levels interact synergistically with temperature to influence transmission dynamics. Cold winter air typically has low absolute humidity, which promotes the formation of smaller aerosol particles that remain suspended in air for extended periods. These smaller particles (<5 μm) can penetrate deeper into the respiratory tract when inhaled [78]. Research indicates that at 40–50% relative humidity—typical of heated winter barns—airborne viruses demonstrate peak stability and infectivity [79,80]. This creates an especially dangerous scenario in modern confined animal housing where recirculated air may contain high viral loads. Housing conditions during winter months further exacerbate these risks. Producers typically implement barn closures and reduced ventilation to conserve heat, inadvertently creating ideal conditions for viral accumulation. Studies of swine production facilities demonstrate that airborne virus concentrations can be 3–5 times higher in winter housing configurations compared to well-ventilated summer conditions [81]. Animal crowding, a common practice during cold weather to utilize body heat, dramatically increases contact rates and facilitates rapid viral spread through populations. The combination of these factors explains why winter outbreaks often affect entire barns within days, while summer transmission tends to be more sporadic.

### 5.2. Host and Viral Population Factors

Host population characteristics interact with these environmental conditions to create the perfect conditions for seasonal epidemics. Immunity patterns fluctuate annually in livestock populations due to several key factors. Most production systems implement vaccination programs in late fall, creating population-level immunity that wanes progressively through the winter months. This is compounded by the continuous introduction of new, immunologically naïve animals through standard production cycles. For example, in swine operations, the regular introduction of replacement gilts and weaned piglets creates predictable immunity gaps that viruses exploit [82]. Age-dependent susceptibility represents another critical factor. Young animals are particularly vulnerable due to immature immune systems and lack of prior exposure. In PEDV outbreaks, mortality rates approach 100% in neonatal piglets compared to minimal clinical signs in adults [52]. Cold stress may further compromise immunity in young stock through cortisol-mediated suppression of lymphocyte function and reduced mucosal antibody production.

The viruses themselves undergo seasonal evolutionary pressures. The large, immunologically naïve populations present in winter create ideal conditions for viral mutation and selection. Antigenic drift occurs more rapidly under these conditions, allowing viruses to escape existing herd immunity. Recombination events between coinfecting strains are also more common in winter populations, potentially giving rise to novel variants [83,84]. These evolutionary processes help explain why winter outbreaks often involve emerging strains against which existing vaccines may be less effective. Management practices inadvertently contribute to these patterns. The common practice of holding animals in larger groups during winter increases viral replication opportunities. Transportation of animals between farms during fall months facilitates viral spread before winter confinement begins. Even feeding practices change in winter, with cold-weather rations sometimes containing different ingredients that may affect gut immunity and microbiome composition [85]. Together, these factors create a perfect storm of conditions that explain the striking seasonality of viral diseases in temperate livestock production systems. Understanding these complex interactions is essential for developing more effective prevention and control strategies.

## 6. Conclusions

This comprehensive review synthesizes current knowledge on the pivotal role of HSP70 in mediating the relationship between cold stress and seasonal viral diseases in farm animals. We have demonstrated that cold-induced HSP70 overexpression serves as a double-edged sword—while it enhances cellular protection against thermal stress, it simultaneously facilitates viral replication through chaperone functions and modulates immune responses in ways that may increase host susceptibility. The evidence presented reveals HSP70’s involvement across multiple viral life cycles, including PEDV, PRRSV, and BVDV, through mechanisms ranging from viral protein stabilization to replication complex assembly. Importantly, we have contextualized these molecular interactions within the broader framework of winter disease epidemiology, acknowledging the concurrent roles of environmental factors (temperature and humidity), husbandry conditions, and viral evolution in driving seasonal patterns. The findings underscore the need for a paradigm shift in how we approach winter disease management, moving beyond traditional biosecurity measures to incorporate stress physiology and molecular pathways.

## 7. Future Perspectives

The investigation of HSP70’s role in cold-stress responses and viral disease seasonality opens numerous avenues for future research with significant implications for livestock health management. Building on current findings, several critical directions emerge that warrant focused investigation. First, comprehensive field studies are needed to establish quantitative relationships between HSP70 expression levels, environmental stress parameters, and viral disease incidence across different livestock production systems. Such studies should incorporate advanced monitoring technologies, including wearable sensors for real-time stress assessment and automated environmental data collection systems, to capture the complex interactions between animal physiology and husbandry conditions. Second, the development of targeted HSP70 modulation strategies presents a promising intervention approach. This includes screening natural compounds (e.g., plant-derived polyphenols and probiotics) for their ability to fine-tune HSP70 expression without compromising essential chaperone functions. Parallel efforts should explore pharmacological inhibitors specifically designed to disrupt proviral HSP70–virus interactions while preserving its cytoprotective roles. The potential for nutritional programming, where early-life dietary interventions induce long-term HSP70 regulation patterns, represents another innovative avenue worth exploring. Third, genetic and epigenetic research should investigate breed-specific differences in HSP70 induction patterns and their correlation with cold resilience. Advanced genomic tools, including genome-wide association studies and CRISPR-based gene editing, could identify genetic markers for selective breeding programs aimed at optimizing stress responses. Similarly, research into epigenetic modifications induced by chronic or intermittent cold exposure may reveal novel regulatory mechanisms of HSP70 expression. Fourth, the interplay between HSP70 and gut–immune axis signaling requires deeper exploration. Particular attention should be given to how cold-stress-induced HSP70 affects gut barrier integrity, microbiota composition, and mucosal immunity—all critical factors in enteric viral infections. This research should employ multi-omics approaches to unravel the complex networks connecting molecular chaperones, microbial communities, and immune function. Fifth, climate change adaptation strategies must incorporate findings on HSP70 dynamics. Predictive modeling that integrates HSP70 expression data with climate projections could help anticipate shifts in disease patterns and inform the development of region-specific management protocols. This is particularly relevant for temperate regions experiencing increasing climate variability.

Finally, translational applications should focus on developing practical intervention packages that combine HSP70-targeted approaches with existing biosecurity measures. This includes optimizing barn insulation and ventilation systems, refining cold acclimation protocols, and designing seasonal nutritional supplements—all while maintaining rigorous cost–benefit analyses to ensure economic viability for producers. Implementation of these research directions will require strong interdisciplinary collaboration among virologists, physiologists, geneticists, climatologists, and livestock production specialists. The ultimate goal is to transform our understanding of HSP70 biology into actionable strategies that enhance animal welfare, improve productivity, and increase the resilience of livestock systems facing changing environmental challenges. By addressing these knowledge gaps, we can develop more precise and effective approaches to mitigate the impacts of seasonal viral diseases in farm animals, contributing to global food security and sustainable animal production.

## Figures and Tables

**Figure 1 microorganisms-13-01755-f001:**
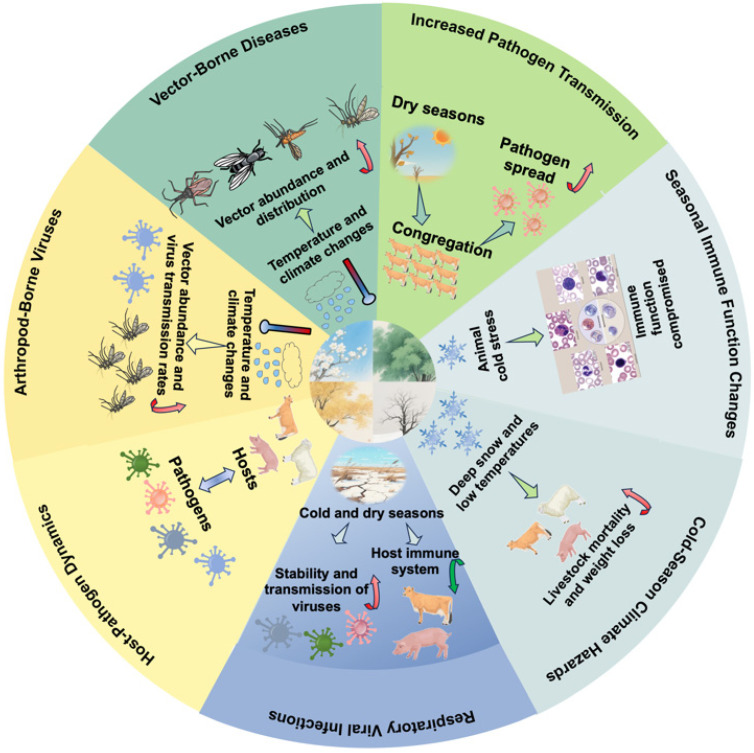
Multifactorial drivers of viral disease seasonality in intensive animal production systems. This schematic illustrates the complex interplay of environmental drivers (temperature and humidity), host-related factors (animal behavior, physiological stress, and immune function), pathogen characteristics (viral stability and transmission routes), vector dynamics, and husbandry practices that collectively influence the seasonal patterns of viral diseases in animal husbandry. The central circle highlights the impact of cold-season climate hazards on livestock, emphasizing the role of immune compromise. Arrows indicate the direction of influence between factors.

**Figure 2 microorganisms-13-01755-f002:**
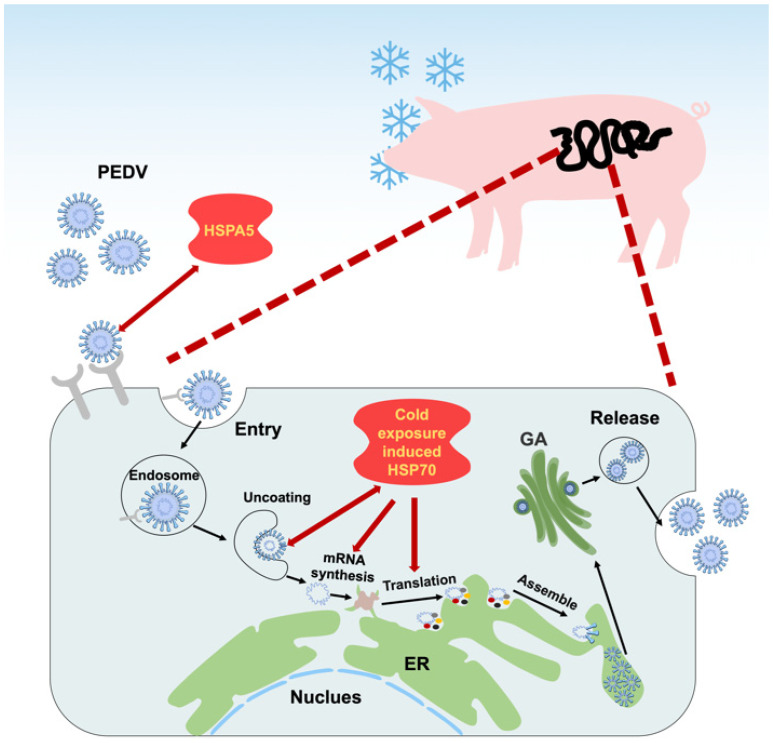
Roles of HSP70 at various steps in the PEDV life cycle. This diagram depicts the PEDV life cycle within a host cell, highlighting the involvement of host HSP70 at different stages. Double-sided red arrows denote binding interactions between host HSP70 and the PEDV spike (S) or membrane (M) proteins. Single-sided red arrows indicate key steps mediated or facilitated by HSP70 activity, including viral entry, uncoating, mRNA synthesis, and translation. Cold-exposure-induced HSP70 upregulation is shown to influence these processes.

## Data Availability

No new data were created or analyzed in this study. Data sharing is not applicable to this article.

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
