# Peer review of "Heat Shock Protein 70 in Cold-Stressed Farm Animals: Implications for Viral Disease Seasonality"

_microorganisms, 2025, doi:10.3390/microorganisms13081755_

Round 1
Reviewer 1 Report
Comments and Suggestions for Authors
Kong et al.'s mini-review, "Heat Shock Protein 70 in Cold-Stressed Farm Animals: Implications for Viral Disease Seasonality," is a clear, relevant, and well-structured contribution addressing a timely and crucial topic in veterinary virology and immunophysiology. It investigates the link between cold stress in livestock, Heat Shock Protein 70 (HSP70) expression, and the seasonal patterns of viral disease outbreaks. The review comprehensively highlights that cold stress significantly upregulates HSP70 in various farm animal organs, where it exhibits a dual function: protecting host cells while also facilitating viral infections. Across poultry, cattle, quail, and pigs, this elevated HSP70 expression in cold conditions not only improves cold tolerance but is also associated with enhanced replication of viruses such as PEDV, PRRSV, and BVDV. The seasonal prevalence of these viral diseases, especially during winter and early spring, is attributed to cold-induced immune suppression and increased viral stability. Mechanistically, HSP70 aids viral replication by stabilizing viral complexes and facilitating their entry and assembly. Furthermore, it modulates immune responses, exhibiting both immunosuppressive effects (e.g., promoting IL-10 and TGF-β production) and immunostimulatory effects (e.g., influencing IFN signaling and antigen presentation), depending on the specific context. The review concludes that HSP70 acts as a crucial molecular link between environmental cold stress and seasonal viral outbreaks in livestock, suggesting its potential as a biomarker and therapeutic target. It also calls for further research into species-specific responses, gut microbiota interactions, and selective breeding strategies.
The manuscript is well-written, logically organized, and appropriately scoped as a mini-review, providing a focused synthesis rather than a comprehensive systematic review. Its introduction effectively frames the core issue, and subsequent sections—covering HSP70 expression dynamics, seasonal viral trends, mechanistic roles of HSP70, and conclusions—are coherently structured. Most cited references (2018–2024) are recent, scientifically relevant, and feature minimal self-citations, primarily supporting mechanistic claims regarding HSP70’s role in stress response, viral replication, and immune modulation, with specific examples including its role in PEDV replication regulation (lines 127–128) and viral chaperoning (lines 163–173). While the review lacks original data or a formal methodology section, its literature synthesis appears rigorous; however, a brief explanation of the study selection process would enhance transparency. Figures and tables, such as Table 1 (summarizing HSP70 expression across species and organs) and Figure 2 (illustrating HSP70's role in the PEDV lifecycle), are generally appropriate. Nevertheless, standardizing Table 1's formatting and adding a figure to illustrate immune modulation would further strengthen the paper. Data interpretation consistently remains cautious, avoiding overstated conclusions. The conclusions align with the cited evidence and propose valuable future directions, including the investigation of HSP70 as a biomarker or therapeutic target (lines 231–240). Ethics and data availability declarations are adequate for a review, with funding and conflict of interest statements clearly provided (lines 245–247). The review successfully identifies a significant knowledge gap concerning the environmental modulation of viral pathogenesis in farm animals. It offers a novel synthesis by linking HSP70 expression to seasonal disease outbreaks, an area not extensively explored in recent literature. Overall, the review maintains strong coherence in its argumentation and scientific support, with minor exceptions such as an unsubstantiated mention of "unpublished data," which should either be elaborated upon or removed for enhanced rigor.
Reviewer 2 Report
Comments and Suggestions for Authors
In this manuscript “Heat Shock Protein 70 in Cold-Stressed Farm Animals: Implications for Viral Disease Seasonality” by Kong et al., the authors have reviewed existing literature on HSP70, cold-stress response, role of HSP70 in viral-replication and anti-viral immune response. Understanding cold-stress response in farm animals is critical to develop strategies to mitigate any potential impact on overall animal welfare (animal weight gain, reproduction, etc) thus leading to economical loss. Specific comments.
- Since cold-stress induces a multi-factor response, it would be better to focus impact of cold-stress induced changes in overall farm animal-welfare i.e. animal weight gain, reproduction rate, litter size, feed consumption etc. It is very difficult to justify cold-stress induced HSP70 is the only critical factor.
- Positive or negative regulation of virus replication by HSP protein are usually based on in-vitro cell culture models, therefore it is unwarranted to speculate changes in HSP70 levels attributes to winter seasonality of viral disease in farm animals. Environmental factors (low temp, relative humidity, over-crowding, aerosol size, etc) affect virus persistence in a given environment, and subsequent transmission amongst a given population. Additionally, host factors such as age, herd-immunity status, along with viral factors such as antigenic drift/shift plays a major role in disease severity and seasonality. Therefore, it would be better to focus how cold environment is conducive for viral persistence and other factors involved disease seasonality amongst farm animals during winter.
